# Education for Healthcare Providers: Impact of Academic Detailing on Reducing Misinformation and Strengthening Influenza Vaccine Recommendations

**DOI:** 10.3390/pharmacy12060188

**Published:** 2024-12-23

**Authors:** Kimberly C. McKeirnan, Megan E. Giruzzi, Damianne C. Brand, Nick R. Giruzzi, Kavya Vaitla, Juliet Dang

**Affiliations:** 1Pharmacotherapy Department, Washington State University College of Pharmacy and Pharmaceutical Sciences, Yakima, WA 98901, USA; megan.giruzzi@wsu.edu (M.E.G.); dbrand@wsu.edu (D.C.B.); nicholas.giruzzi@wsu.edu (N.R.G.); kvaitla2@gmail.com (K.V.); 2CSL Seqirus, Summit, NJ 07901, USA; juliet.dang@seqirus.com

**Keywords:** influenza vaccination, academic detailing, healthcare provider education, vaccination barriers

## Abstract

Background: Recommendations from a trusted healthcare provider have been shown to be the most effective intervention for encouraging patients to be vaccinated. However, providers have reported feeling less prepared to address vaccination questions and having less time to discuss vaccines with patients than before the COVID-19 pandemic. Providers may benefit from a brief update about the available influenza vaccines and vaccination guidelines. Academic detailing is an evidence-based approach for preparing healthcare providers to discuss getting vaccinated with patients. Methods: An academic detailing presentation was developed using influenza statistics, vaccination recommendations, and recent local and national immunization rate data. Academic detailing was conducted with physicians and community pharmacy personnel in Yakima County, Washington, between November 2023 and January 2024. Yakima County is designated as a medically underserved area due to a lack of providers. A pre-detailing survey was conducted to evaluate participant knowledge of current ACIP recommendations and gather opinions about local resident vaccination barriers. A post-detailing survey was conducted to gather participants’ opinions about the value of detailing. Results: Prior to the training, 73% of providers believed it was important to discuss influenza vaccination with patients, but only 52% felt confident in combating misinformation. Healthcare providers believed misinformation and vaccine hesitancy are the most common barriers for Yakima County patients, but recent survey results showed that online scheduling systems, long wait times, and limited appointment hours were the predominant issues reported locally. Two out of 12 community pharmacy personnel and zero resident physicians correctly named all three preferentially recommended influenza vaccines for patients 65 years and older. Overall, 96% of detailing participants reported that the session was valuable, 87% believed it would help them combat vaccine misinformation, and 65% reported planning to have more conversations with patients about influenza vaccination after participating. Conclusion: Physicians and community pharmacy immunizers found the influenza vaccines academic detailing to be valuable. Staying up to date on vaccination guidelines can prepare providers to be confident in having informed conversations with patients about getting vaccinated.

## 1. Introduction

Influenza continues to have a significant impact on the healthcare system, leading to 100,000–710,000 hospitalizations and 49,000–51,000 deaths annually between 2010 and 2023, with an overall cumulative end-of-season hospitalization rate of 66.2 per 100,000 in 2019–2020 [1]. In February 2024, the Centers for Disease Control and Prevention (CDC) reported estimates of 280,000–580,000 hospitalizations and 17,000–50,000 deaths secondary to the flu for the 2023–2024 U.S. flu season [2]. The 2010–2011 and 2022–2023 end-of-season influenza vaccination rates indicated that 42–52% of the population over the age of six months received at least one influenza vaccination [3]. The Advisory Committee on Immunization Practice (ACIP) recommends that all persons aged ≥ 6 months, who do not have contraindications, receive a seasonal influenza vaccine [4]. Prior to age 65, there is no preferred recommendation among the different influenza vaccinations with the exception of solid organ transplant recipients, but the ACIP preferentially recommends using one of three vaccines for adults 65 years and older: trivalent high-dose inactivated influenza vaccine (HD-IIV4), trivalent recombinant influenza vaccine (RIV4), and trivalent adjuvanted inactivated influenza vaccine (aIIV4). In 2023, these recommendations were similar but recommended the quadrivalent vaccine instead of the trivalent influenza vaccine [5].

Provider recommendations for the influenza vaccination are a fundamental part of increasing the yearly vaccination rates. The data suggests provider recommendations are the single most effective intervention in whether or not a patient will be vaccinated [6]. A systematic review and meta-analysis composed of 840,838 patients across 15 counties found that physician recommendation (r = 0.46 (95% CI 0.34 to 0.56)) had the greatest influence on parents’ uptake of the human papillomavirus (HPV) vaccines for their children [6]. Although provider recommendations play an integral role in patient vaccination rates, the COVID-19 pandemic has led to increased barriers during patient-provider interactions. Providers report being faced with more questions than in previous years, feeling ill-prepared to field the questions, and being stretched thin for time [7]. To overcome these challenges, providers may benefit from a refresher on influenza, the influenza vaccines available, the ACIP influenza vaccine updates, motivational interviewing techniques, and the impact of healthcare providers on patients’ vaccination decision making.

One method that has shown previous success in improving provider knowledge about vaccines and indirectly enhancing patient education is academic detailing. Academic detailing is a type of interactive educational outreach conducted by a trained healthcare professional [8]. It consists of face-to-face discussions in a one-to-one or group setting. The key to successful academic detailing is keeping the provider engaged and ending the visit with an agreed-upon commitment to specific practice changes [8,9]. Academic detailing emphasizes first performing an individualized needs assessment, then utilizing compelling educational “Detailing Aids” to illustrate a problem, and lastly facilitating evidence-based discussions to overcome the identified problems and encourage behavioral change. Academic detailing provides a safe environment to establish a rapport, stimulate shared decision making with providers, and improve patient care [10].

Academic detailing has been utilized in various healthcare settings to improve patient care [11]. One way that academic detailing has been implemented is as an intervention to help improve prescribing patterns. One study that focused on the prescribing patterns of medications related to type II diabetes and hypertension found that when providers received academic detailing, it led to an appropriate increase in the use of metformin for diabetes, from 25.7% to 34.8% (*p* < 0.005), and an appropriate decrease in the use of beta blockers for blood pressure, from 17.9% to 14.5% (*p* < 0.005) [12]. Similarly, academic detailing was associated with a 92% reduction in antibiotic prescribing for asymptomatic bacteremia [13]. These studies demonstrate how academic detailing improves the quality of care.

Additionally, academic detailing has been utilized to improve immunization rates. One pharmacist-led academic detailing team developed a statewide program (at 400 sites) that included a decision making support pathway, vaccination schedule, and corresponding indications for the pneumococcal vaccine [14]. Statewide pneumococcal vaccinations increased from 72.4% in 2013 to 76.3% in 2015 (*p* = 0.01). The study also found a decrease in pneumococcal disease, hospitalizations, and mortality following the intervention [14]. Similarly, a student-led academic detailing used to provide educational outreach on the pneumococcal vaccine to community pharmacies was found to be beneficial [15]. The overwhelming majority of community pharmacists (93.8%, 61/65) were confident that they could apply the knowledge that they obtained in their clinical practice and intended to utilize the vaccination pathway that was presented in their clinical practice [15].

The literature suggests that academic detailing can improve the appropriate use of vaccinations, vaccination rates, and subsequently improve associated morbidity and mortality [11,12,13,14]. The objective of this work is to describe the implementation of academic detailing about influenza vaccines for community pharmacists and physicians and to gather participants’ opinions about the value of the detailing.

## 2. Materials and Methods

### 2.1. Academic Detailing Preparation

Prior to the 2023–2024 influenza season, an influenza survey was sent out to the residents of Yakima County. This survey identified a gap in provider recommendations for the influenza vaccine [16]. The survey found that only 74.7% of patients reported a healthcare provider recommending a vaccination against the flu within the last five years, in 2018–2022. The vast majority, 95.4% of those who responded being recommended the vaccine, were recommended the vaccine by their primary care provider (PCP). When asked specifically about pharmacists, only 38.1% of the residents reported a pharmacist recommending the vaccine in the last five years, and when asked about non-PCP providers (e.g., cardiologists, pulmonologists) recommending the vaccine, only 27.2% answered yes. This information was utilized to develop an academic detailing presentation for community pharmacists and medical providers that serve the community where the patients who participated in the survey reside.

In preparation to offer academic detailing, four members of the project team participated in academic detailing skills training through the National Resource Center for Academic Detailing (NaRCAD) [17]. The fifth member of the project team had previously attended this training in 2015. The NaRCAD 3-day virtual academic detailing training provided the framework and tools necessary for educators to conduct successful academic detailing sessions to promote exceptional patient-centered care. A brief academic detailing presentation was developed by the project team using influenza incidence and mortality rates from the CDC [1,2] and Washington State Department of Health [18], as well as the ACIP influenza immunization recommendations [5] and results from an immunization survey tailored towards Yakima County residents [16]. The principles learned during the NaRCAD training were applied to the creation of the detailing material [9,17].

### 2.2. Study Location

Academic detailing was provided to physicians and community pharmacy personnel in Yakima County, Washington. Yakima County has been designated by the Health Resources and Services Administration as a medically underserved area, defined as having a shortage of primary care health services for residents within a geographic area [19]. Yakima County has approximately 250,873 residents, and the city of Yakima has an estimated 94,000 residents [20].

### 2.3. Study Participants

The outpatient clinic site for academic detailing consists of over 60 providers and is also the hub for the community’s only family medicine residency, which consists of 24 residents in the three-year program. Onsite medical residents train to become proficient in all aspects of both outpatient and inpatient family medicine. Furthermore, these providers receive the majority of their inpatient training at MultiCare Yakima Memorial Hospital (MYMH) [21]. MYMH is a 226-bed acute-care, not-for-profit community hospital that has served Central Washington’s Yakima Valley since 1950. MYMH includes a multispecialty team of more than 300 practitioners and 20-plus primary care and specialty care locations. Two researchers involved in the study are employed by Washington State University College of Pharmacy and Pharmaceutical Sciences (WSU CPPS) [22] to provide experiential education to student pharmacists and physicians through the MYMH family medicine residency training program; thus, the outpatient clinic was selected for academic detailing. The team of providers at MYMH was ideal for detailing about the influenza vaccine, as they work not only at the clinic but also rotate through the only hospital in Yakima, as mentioned above.

Opportunities for detailing community pharmacists and technician vaccinators were identified through the compilation of a list of all community pharmacies located in Yakima County using a licensed facility search tool from the Washington State Department of Health [23]. In Yakima County, encompassing the towns of Yakima, Wapato, Toppenish, Grandview, and Selah, there are currently more than 40 pharmacies serving approximately 250,000 residents. These include large chain stores, clinic-based pharmacies, and independent pharmacies. This list was reduced to within 10 miles from the WSU CPPS Yakima campus to allow for accessibility to the researcher.

### 2.4. Provision of Academic Detailing

For the physicians, detailing was provided in a group setting rather than in a one-on-one setting. The ideal time for academic detailing was determined to be during the weekly didactic sessions, where residents gather to explore a variety of topics. Residency didactic sessions are conducted both in person and virtually, as many residents are off-site completing different rotations. To minimize the risk of burnout, detailing was conducted early in these sessions.

The community pharmacies were randomized and chosen from the list described above. A researcher contacted the pharmacy manager during normal business hours to request time for a 15 min detailing appointment. The managing pharmacist was also encouraged to include another of their trained immunizing staff to join the detailing meeting. The sites varied in size, the volume of prescriptions and vaccinations they delivered, and their ownership (chain pharmacy vs. independently owned). Of the pharmacies visited, all pharmacists were trained as immunizers, and many had technicians also trained as immunizers. Each visit lasted approximately 15 min to avoid negatively impacting pharmacy workflow. Several pharmacies had one other pharmacist to maintain workflow during their absence. Two pharmacies only had one pharmacist working during the detailing meeting, so care was taken to ensure the session took place while they were also present to supervise pharmacy workflow.

### 2.5. Evaluation of Academic Detailing

Two surveys were developed for the academic detailing session: a pre-academic detailing survey and a post-academic detailing survey. The pre-academic detailing survey had several purposes: to engage participants in thinking about current influenza vaccination practices at their practice site; to gather participant demographics; to briefly evaluate participant knowledge of current ACIP recommendations regarding influenza vaccines [5]; and to gather participant opinions about vaccination barriers and Yakima County patient behaviors. The post-academic detailing survey was designed to gather participants’ opinions about the value of the influenza vaccine academic detailing and to identify topics for future academic detailing discussions. The survey tools are shown in Appendix A and Appendix B.

For in-person academic detailing sessions, surveys were conducted in person on paper immediately before and after the academic detailing presentations. Participants were handed a copy of the survey before the academic detailing presentation began and were instructed to complete only the front page, which contained the pre-detailing survey. Participants were given five minutes to complete the survey before the detailing presentation began. Upon completion of the detailing presentation, participants were invited to take the post-detailing survey, which was on the back side of the printed page that participants were given at the beginning.

The methods used in this study were determined to satisfy the criteria for exempt research by the Institutional Review Board (#20194-001). Content at the top of the pre-academic detailing survey questionnaire informed participants about the purpose of the survey; that participation was voluntary and they were free to skip any individual questions or end the survey at any time; that responses would remain anonymous; and of their rights as a study participant and the contact information of the study primary investigator in accordance with the requirements of the Human Research Protection Program at Washington State University.

## 3. Results

Academic detailing was offered between November 2023 and January 2024. Academic detailing sessions were provided to 12 pharmacy immunizers and 11 physicians. Detailing sessions lasted between 10 and 15 min for pharmacists and 15 min for medical residents. Of the academic detailing participants, 100% (*n* = 23) completed the survey. Demographic information is included in Table 1.

All of the physicians reported working in both the clinic and hospital, which is normal practice at this site. Pharmacy personnel were asked how many influenza vaccines are given in an average week during the influenza season. Two of the pharmacies reported administering more than 200 influenza vaccines per week, four reported giving between 50 and 100 per week, and five reported giving less than 50 per week.

Prior to the detailing presentation, participants were asked to name the three influenza vaccines that are preferentially recommended for adults 65 years and older by the ACIP [5]. Only two out of 12 community pharmacy personnel and none of the physicians were able to correctly name all three vaccines, as shown in Table 2. Ten out of the 23 participants correctly named high-dose influenza vaccine, but only five and four participants identified adjuvanted influenza vaccine and recombinant influenza vaccine, respectively.

Next, academic detailing participants were asked which barriers they believe would be reported by people living in Yakima County. The barriers reported by the academic detailing participants are shown in Figure 1, with the most common being vaccine hesitancy or mistrust (30%) followed by vaccine cost and/or lack of insurance (22%). The barriers reported by patients are shown in Figure 2 and include scheduling barriers (long wait times, difficulty using online scheduling tools, lack of appointment availability). Asking academic detailing participants these questions and then revealing the answers reported by patients in their community was intended to provide insight into the patient experience and encourage discussion during the detailing sessions.

Prior to the academic detailing training, participants were asked to rank three questions on a corresponding five-point Likert scale. The questions included: “How confident are you that your strong recommendation to be vaccinated against influenza is important to your patients?”; “How confident are you in combatting misinformation about the influenza vaccine with your patients?”; and “How important is it to you to discuss annual influenza vaccination with your patients?”. The results of these questions can be found in Figure 3.

Upon completion of the academic detailing session, participants were asked to complete a short nine-question survey about their experience, the value of the information provided in the academic detailing session, and their intentions about future conversations with their patients about influenza. Results from the five Likert scale questions are shown in Figure 4.

Participants were also asked whether they intended to have more, less, or the same number of conversations with patients about receiving an annual influenza vaccine after participating in the academic detailing presentation. The majority (*n* = 15, 65%) of respondents reported that they intended to have more conversations with their patients, while 35% (*n* = 8) said they would have the same frequency of conversations with patients about influenza vaccination.

Academic detailing participants were asked to list the most valuable thing that they learned from the presentation. Forty-three percent of participants reported that they were surprised by the data showing that patients were not as against vaccines as expected and intended to place more importance on discussing vaccinations with patients. One physician stated, “I need to do better”. Thirty percent reported that the most valuable information from the session was learning about the lack of influenza vaccinations occurring in pregnant patients and discussing how to address this. Seventeen percent reported that the most important information was the discussion about the three preferentially recommended influenza vaccines for patients 65 years and older.

## 4. Discussion

The results of this academic detailing project identified a worrisome knowledge gap among healthcare providers regarding ACIP recommendations for influenza vaccine use in adults 65 years and older. Two out of 23 providers were able to correctly name the three preferentially recommended influenza vaccines. This is particularly problematic since previous research suggests that provider recommendations for the influenza vaccination are the single most effective intervention in encouraging patients to be vaccinated [6]. However, 96% of healthcare providers in this study reported that the information provided in the academic detailing session about the preferential recommendations for the influenza vaccine was valuable to them, and 65% reported planning to have more conversations with their patients regarding influenza vaccine after participating in the academic detailing session. Prior to the training, most providers (73%) believed it was very or fairly important to have these annual discussions concerning flu vaccination with their patients, but less (52%) felt very or fairly confident in combating misinformation. After the training, the majority of participants believed the information in the presentation was valuable in combating misinformation (87%). These findings were reflective of other academic detailing successes where education was provided with supporting evidence, either by guidelines or patient perspectives [15,24]. Ninety-one percent of participants reported that they believed the information presented about making a strong recommendation was valuable to them as healthcare providers. A 2023 review identified a negative attitude toward healthcare as the predominant barrier to being vaccinated against influenza, but trust in healthcare services as the strongest promoting factor [25].

Physicians were less knowledgeable about the influenza vaccination recommendations than the pharmacy-based immunizers. This may be related to workflow and delegation of tasks. In the clinic, residents order vaccines through an electronic medical record (EMR). Vaccine options are provided in the EMR based on current availability. After the physician orders the vaccine, a medical assistant or nurse provides the vaccination. The pharmacy-based immunizers are likely more involved in vaccination administration and ordering the supply of vaccines, so they may be more knowledgeable about available options.

This research also highlighted misconceptions among providers about influenza vaccination barriers for their patients. Providers’ opinions about barriers faced by local patients were not aligned with the barriers reported by patients themselves in a recent survey. When asked what they believed were the biggest barriers to influenza vaccination in Yakima County, healthcare providers reported vaccine hesitancy and mistrust (30%) followed by cost or lack of insurance (22%), lack of knowledge about where to receive the vaccine (12%), and challenges with scheduling and wait times (12%) as the predominant barriers. Conversely, a recent survey among Yakima County residents indicated that 50% of the more than 200 respondents reported issues with scheduling, such as difficulty using an online appointment system or limited appointment hours and long wait times, as the predominant barrier. Only 17% of patients reported vaccine hesitancy or mistrust and 7% reported cost or lack of insurance as barriers. Many healthcare providers are still combating COVID-related mistrust with their patients, which may contribute to the higher proportion of providers believing mistrust was a barrier compared with their patients.

Previous studies have identified a negative attitude toward healthcare, in general, as the leading barrier to being vaccinated against influenza [25,26]. When the patients’ reported barriers were shared with providers during academic detailing, there was some surprise and reflection on what could be done to mitigate these barriers through better communication and addressing opportunities to increase access. Understanding the needs and barriers that are specific to local patients could help healthcare providers be prepared to better serve their community and mitigate current biases associated with vaccination barriers.

The results of this research highlight opportunities for actionable practice change that could positively impact immunization uptake. First, ensuring that healthcare providers are up to date on vaccination guidelines could improve both provider knowledge and confidence having conversations with patients about vaccinations. Since previous research reported providers’ lack of confidence answering patient questions as a barrier to vaccination uptake [7], and making strong recommendations has been shown to be among the most effective actions healthcare providers can take to encourage vaccination [6], arming providers with more information could empower them to feel confident initiating these important conversations.

Next, healthcare providers can reevaluate vaccination scheduling systems with the goal of simplifying user interfaces and ensuring vaccinations are available during convenient hours and with minimal wait times. This should also include providing adequate staffing for vaccination services during times that are convenient for patients. Although pharmacies have demonstrated success implementing vaccination services because of being widely accessible with convenient hours [27,28,29], time constraints and workload concerns have been shown to be limiting factors [30]. Providers in this study were surprised to learn that the majority of patients listed difficulty using the online scheduling systems or being available during vaccination hours of operation as their primary barriers. Having healthcare providers and medical staff who are more aware of patient needs and the seriousness of scheduling difficulties could improve vaccination uptake.

Lastly, healthcare providers can expand efforts to build a trusting rapport with patients. Patient trust is an asset in combatting misinformation. Healthcare providers continue to be the most trusted source for patient vaccination information when they have a strong relationship with patients [6,31]. Recent research suggests that demonstrating competence, benevolence, and empathy to patients can promote trustful patient-physician relationships [32], and factors predictive of pharmacist-patient trust include considering patients’ goals, needs, values, and preferences [33].

There are limitations to this work. This was a small study designed to provide academic detailing to a specific group of providers in response to results from a local survey about influenza vaccine, limiting generalizability. This work evaluated provider perceptions about the value of academic detailing, but it was not within the scope of this study to evaluate actual changes in vaccination practices. Additional research evaluating practice change after academic detailing, rather than just provider intentions, could offer more valuable insight and assist in refining academic detailing presentations.

This study was also conducted in one county in Washington State. In Yakima County, 52% of the residents identify as Hispanic or Latino/a, which is much higher than average in Washington State (14%) and the U.S. (19%) [17]. Vaccination rates in the U.S. are typically lower among minority groups than other populations [34,35]. The majority of Yakima County is also considered rural [19]. Rural areas in the U.S. typically have lower average vaccination rates than their urban counterparts [36]. Larger studies representing an expanded geographic area and more diverse demographics would lead to more generalizable results. There may also be response bias associated with survey completion. Previous research has identified that survey respondents may be more likely to provide answers they believe the researchers find desirable [37].

Future projects that involve pharmacists leading academic detailing efforts could have a substantial impact on immunization rates. Although this work was limited to the influenza vaccine, academic detailing has been shown to have positive impacts on vaccination rates for other vaccines, including pneumococcal vaccines and HPV. Studies could also be designed to build on the results of this work by implementing academic detailing for COVID-19 and RSV vaccines. Additionally, incorporating local patient opinions about vaccination barriers into academic detailing can provide insight that is valuable to providers in combating misinformation and empower them to have more conversations with patients about getting vaccinated.

## 5. Conclusions

Academic detailing programs continue to demonstrate success in improving vaccination rates. Physicians and community pharmacy immunizers found the influenza vaccination academic detailing to be valuable and enhanced their knowledge on the current ACIP recommendations. Staying up to date on vaccination guidelines can prepare providers to be confident in having informed conversations with patients about getting vaccinated.

Healthcare providers’ opinions about barriers faced by local patients were not aligned with actual barriers reported by patients. Providers believed misinformation and vaccine hesitancy were the most common barriers in their community, but local patients reported difficulty with scheduling and using online systems, as well as long wait times and limited appointment hours, were the predominant issues. Better understanding of the challenges faced by patients and barriers to accessing vaccinations can improve provider preparedness to serve their community and mitigate current biases associated with vaccination barriers.

## Figures and Tables

**Figure 1 pharmacy-12-00188-f001:**
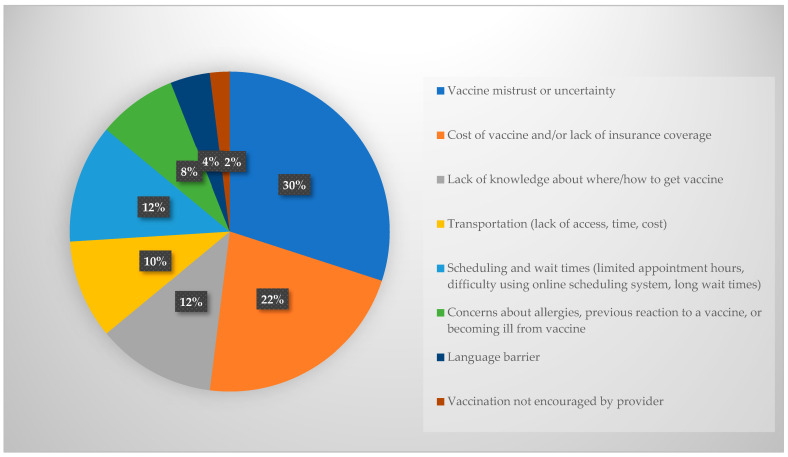
Healthcare provider beliefs about patient barriers to influenza vaccination in Yakima County.

**Figure 2 pharmacy-12-00188-f002:**
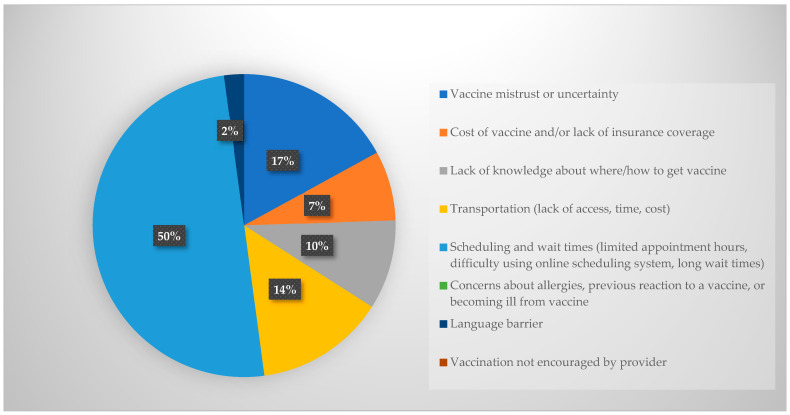
Patient-reported barriers to vaccination in Yakima County.

**Figure 3 pharmacy-12-00188-f003:**
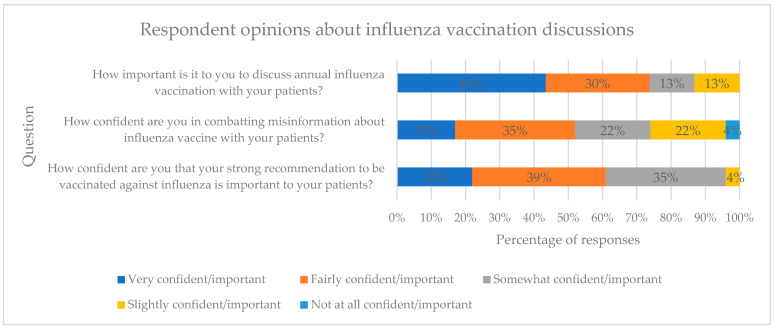
Pre-training survey Likert scale question results.

**Figure 4 pharmacy-12-00188-f004:**
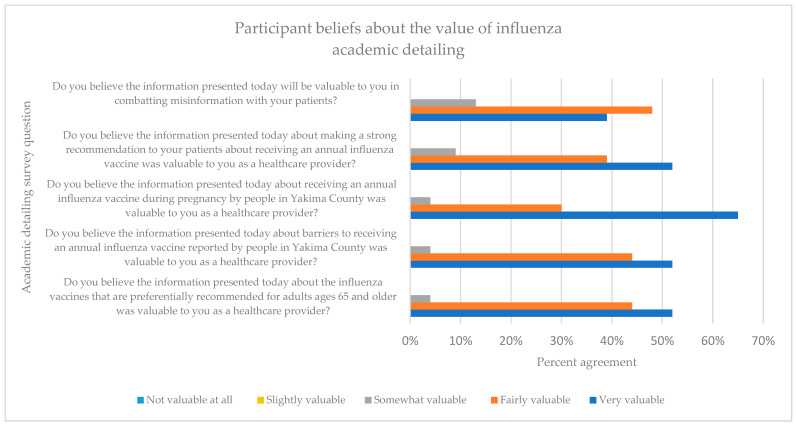
Post-training survey results.

**Table 1 pharmacy-12-00188-t001:** Demographic information for academic detailing participants.

Demographic Question	Role	Number of Respondents (%)
Role	Physician	11 (47.8)
	Pharmacist	8 (34.8)
	Pharmacy Intern	1 (4.3)
	Pharmacy Technician	3 (13)
Years of experience working in the healthcare field	0–5 years	11
	6–10 years	1
	More than 10 years	11

**Table 2 pharmacy-12-00188-t002:** Results of the pre-academic detailing survey question asking healthcare providers to name the three influenza vaccines preferentially recommended for adults 65 years and older by the ACIP [5].

Personnel	Correctly Named One Vaccine	Correctly Named Two Vaccines	Correctly Named Three Vaccines
Community pharmacy personnel	11 out of 12	5 out of 12	2 out of 12
Medical clinic and hospital personnel	2 out of 11	0 out of 11	0 out of 11

* Confidence Likert scale with choices: very confident; fairly confident; somewhat confident; slightly confident; not confident at all. + Importance Likert scale with choices: very important; fairly important; somewhat important; slightly important; not important at all. ^ Value Likert scale with choices: very valuable; fairly valuable; somewhat valuable; slightly valuable; not valuable at all.

## Data Availability

The datasets presented in this article are not readily available because the data are part of an ongoing study. Requests to access the datasets should be directed to the corresponding author.

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
