# Peer review of "Education for Healthcare Providers: Impact of Academic Detailing on Reducing Misinformation and Strengthening Influenza Vaccine Recommendations"

_pharmacy, 2024, doi:10.3390/pharmacy12060188_

Round 1
Reviewer 1 Report
Comments and Suggestions for Authors
1) why only influenza and not covid and rsv?
2)lines 102-104 please provide refs
3)How will the impact of academic detailing on vaccination rates or provider knowledge be measured?
4)What strategies can be implemented to address workflow challenges for pharmacies with limited staff?
5)How can follow-up assessments be used to evaluate and refine detailing sessions?
6) is the sample of the study ok?
Author Response
Thank you for the feedback and the opportunity to revise and resubmit our manuscript. Our responses to your comments are included in the attached document.

Reviewer 2 Report
Comments and Suggestions for Authors
General Comments
This paper investigates the impact of academic detailing on healthcare providers' ability to address misinformation and improve influenza vaccination recommendations. The topic is timely and relevant, particularly regarding increasing vaccine hesitancy and public health challenges. However, several areas need further clarity and expansion to enhance the paper's rigor and relevance.
Major Comments
- Provider recommendations for the influenza vaccination are a fundamental part, and it is suggested that the authors compare more past studies to see how healthcare providers affect patients and how they can increase their trust.
- What are the contributions to the literature? More theoretical and/or empirical support is needed to strengthen the contributions and implications of the research results.
- The study is limited to Yakima County, Washington. While this focus is valuable for localized insights, discussing how findings might generalize to other regions with differing demographics or healthcare infrastructures is recommended. The authors can describe the vaccination rates over the years compared with other areas.
- A key finding is the divergence between providers' perceptions of vaccination barriers and actual patient-reported barriers. Expanding on how this gap could be addressed in future interventions would add value.
- Separating results from implications for practice and future research would improve readability and clarity. For instance, actionable steps for healthcare managers or policymakers could be highlighted distinctly.
Minor Comments
- The results of this academic detailing project identified a worrisome knowledge gap among healthcare providers. This point of view provides good policy implications. It is suggested that the author think about how to let the medical staff who contact the patients understand the seriousness and, simultaneously, think about the best time to intervene and explain.
- In addition, the expression method of the table can also be considered to be displayed from different role perspectives so that subsequent research can continue similar discussions.
Author Response

(The authors gave the same response as above.)

Round 2
Reviewer 1 Report
Comments and Suggestions for Authors
1) Please separete the figure 1 in two
2) Present the surver tool in appenix also
Author Response
Thank you for the feedback, and I apologize for not understanding the request during the first round of revisions.
1) Please separate the figure 1 in two
– Figure 1 separated into two figures as requested. The figures have also been renumbered accordingly.
2) Present the survey tool in appendix also
– the survey tools have been moved to the appendix and are now labeled Appendix A and Appendix B.